# Sleep Quality and Self-Reported Symptoms of Anxiety and Depression Are Associated with Physical Activity in Patients with Severe COPD

**DOI:** 10.3390/ijerph192416804

**Published:** 2022-12-14

**Authors:** Christopher D. Neale, Pernille E. Christensen, Christian Dall, Charlotte Suppli Ulrik, Nina Godtfredsen, Henrik Hansen

**Affiliations:** 1Department of Physical and Occupational Therapy, Copenhagen University Hospital, 2400 Copenhagen, Denmark; 2Department of Quality, Copenhagen University Hospital, 3400 Hillerod, Denmark; 3Institute for Clinical Medicine, University of Copenhagen, 2200 Copenhagen, Denmark; 4Respiratory Research Unit and Department of Respiratory Medicine, Copenhagen University Hospital, 2650 Hvidovre, Denmark

**Keywords:** COPD, physical activity, anxiety, depression, sleep

## Abstract

Sleep quantity, quality and symptoms of depression or anxiety potentially affect the level of daily physical activity (PAL) and plausibly counteracts benefits from pulmonary rehabilitation programs. Their collective impact on PAL is sparsely investigated, particularly in patients with severely progressed chronic obstructive pulmonary disease (COPD). **Aim**: To investigate if sleep quantity, quality and symptoms from self-reported hospital anxiety and depression scores (HADS) are associated with PAL. **Methods**: In this exploratory cross-sectional study data were analysed from 148 participants with COPD; GOLD grade II-IV; GOLD group B to D (52% female, mean 69.7 ± SD of 8.4 years, FEV1% predicted 33.6 ± 10.9, 6MWD 327 ± 122 m, CAT 20 ± 7 points), eligible for conventional outpatient hospital-based pulmonary rehabilitation. Participants had sleep and PAL measured 24 h per day for five consecutive days with an activPAL monitor. Adjusted negative binomial regression was applied to investigate the associations with PAL. **Results**: Participants walked median (25th, 75th percentile) of 2358 (1325.75; 3822.25) steps per day and 14% walked >5000 steps per day on average. Time in bed (TIB) were a median (25th, 75th percentile) of 8.3 (7.1; 9.7) hours and numbers of nocturnal sleeping bouts (NSB) were 1.5 (0.8; 3), Anxiety (HADS-A) and depression (HADS-D) scores were median (25th, 75th percentile) of 5 (3; 8) points and 3 (2; 6) points, respectively, whereof 29% (HADS-A) and 15% (HADS-D) reported scores ≥8 points indicating significant symptoms. The fully adjusted rate ratio (RR) for steps per day for TIB (hours) [RR 0.97 (95% CI: 0.92; 1.02)], NSB (numbers) [RR 1.02 (95% CI: 0.97; 1.07)] were not significantly associated with number of steps per day, while there was a significantly association with number of steps per day for HADS-A [RR 1.04 (95% CI: 1.01; 1.07)] and HADS-D [RR 0.95 (95% CI: 0.91; 0.99)]. **Conclusion**: This exploratory cross-sectional study found a statistically significant association between HADS-A and HADS-D with numbers of steps per day in patients with severe COPD.

## 1. Introduction

Physical activity, defined as any bodily movement produced by the contraction of skeletal muscles that increases energy expenditure [1], is clearly reduced in patients with COPD and constitutes an independent risk factor for mortality, exacerbation, functional mobility loss and further deterioration of other intra- and extra pulmonary manifestations of COPD [2,3]. Importantly, physical inactivity is modifiable. An increase in physical activity has been found in patients with COPD following exercise-based pulmonary rehabilitation (PR) programs or physical activity counselling programs [4,5]. Yet, it is well known that a relatively large proportion of patients neither respond nor maintain a meaningful increase in the level of daily physical activity (PAL) despite being compliant to a provided intervention program [4,5,6,7]. The full picture of both decreased PAL and inability to increase and maintain PAL as such, is far from being known. Existing evidence in patients with COPD have identified degree of airflow obstruction, exercise capacity, respiratory symptoms, muscle- skeletal pain and number of comorbidities as central risk factors for reduced PAL [2,8,9,10,11], meanwhile the association between PAL and anxiety, depression and sleep remain less investigated [3,12,13,14].

A considerable proportion of patients with COPD experience symptoms of anxiety and depression, which is associated with reduced quality of life (QoL), increased risk of hospitalization and increased risk of mortality [15,16,17,18,19,20]. The exact reasons why more patients with COPD manifest elevated rates of anxiety and depression scores remain unclear. The currently available studies strongly associate the severity of individual COPD symptoms with anxiety and depression symptoms [20]. Furthermore anxiety and depression symptoms overlap with physical symptoms of COPD and are thus likely influenced by complex interactions between illness severity, physical capacity and pathophysiological manifestations of COPD in the brain [20,21,22]. The interaction between anxiety and depression symptoms and PAL are most likely bidirectional. A decrease in PAL leads to deconditioning and increases the cardio-respiratory workload during physical activities [2,8]. The increase in cardio-respiratory workload may manifest in shortness of breath and mistakenly be interpretated as disease progression rather than deconditioning and this in turn may lead to periodical or permanent vicious circles of fear avoidance behavior and increase in anxiety and/or depressive symptoms [11,20]. On the other hand, anxiety attacks, for example, may manifest as unexplained sudden shortness of breath or tachypnoea, potentially mimicking the presence of acute exacerbation of COPD, creating fear, panic and restlessness leading to unintentional increase of movement such as frequent body transitions. By contrast, chronic anxiety is the tenacious feeling of anxiety which can interfere with daily life by reduction of daily activities leading to dyspnea [20]. Depression entails affective, cognitive and neurovegetative symptoms, often accompanied by fatigue and psychomotor slowing, which also largely overlap with symptoms of COPD [22]. Thus, depressive symptoms in COPD may amplify reduced PAL and social withdrawal each of which may lead to lower self-confidence, lower self-care poor compliance to treatment and change in physiologic and psychologic sensation [16,22].

Sleep disturbances leading to sleep deficiency is an acknowledged comorbidity in patients with COPD that leads to adverse outcomes including reduced QoL, fatigue and social isolation [23,24,25]. Paradoxically, the presence of sleep disturbances apart from obstructive sleep apnea (OSA) is only scarcely covered in central guidelines [26,27]. For example, the Global Initiative for Chronic Obstructive Lung Disease (GOLD) guidelines and the American Thoracic Society/European Respiratory Society statement for PR only mention sleep in the context of monitoring, while pharmacologic and non-pharmacologic treatment and interactions are not mentioned [26,27]. The relationship between poor sleep and COPD are likely bidirectional and multiple factors contribute to poor sleep such as physiological changes (e.g., hypoxemia), nocturnal bouts (e.g., cough, dyspnea), coexisting sleep disorders (e.g., OSA), psychological disorders (e.g., anxiety/depression) and medication (e.g., corticosteroids, diuretics) [23]. Sleep disturbances increase the risk of developing anxiety and depression two-fold, and the combination of depression and sleep disturbance increases the likelihood of hospitalization in COPD by a factor five [23]. While it is theoretically assumed that increased physical activity will improve sleep disturbances, evidence does not support this assumption [2,3,8,28]. On the contrary, data exploring the association between sleep measures and PAL are sparsely researched and poor sleep may blunt increase of PAL in general thus attenuating the effects on PAL from exercise-based PR programs or physical activity counselling programs [12,13,23].

We are not aware of any studies investigating the association of self-reported anxiety and depressions symptoms and sensor monitored sleep measures with objective measures of PAL in patients with COPD. Thus, the aim of this exploratory retrospective, cross-sectional study was to investigate if sleep quantity, number of nocturnal sleeping bouts and self-reported anxiety and depression are associated with PAL in patients with severe COPD meeting the routine inclusion criteria for out-patient hospital-based PR. Our hypothesis was that higher levels of self-reported anxiety and depression symptoms, excessive time in bed and increased number of sleep interruptions would be associated with fewer steps per day.

## 2. Methods

### 2.1. Study Design and Participants

This exploratory retrospective, cross-sectional study was based on data from two existing clinical cohort studies (NCT02667171 and NCT04249388). Inclusion and exclusion criteria were identical for both cohorts and followed the standard outpatient hospital-based PR in the Capitol Region of Copenhagen, Denmark. Patients were included from six different respiratory departments across hospitals in Greater Copenhagen, from March 2016 to December 2021. The sample consisted of 148 patients with a diagnosis of COPD (FEV_1_/FVC < 0.7), FEV_1_ < 55% (GOLD 2–4), Medical Research Council dyspnea scale ≥ 2 and with no previous participation in any PR within 6 months from baseline and who had worn an activPAL sensor (AP) measuring PAL. Physical activity level measurement was limited to patients who lived within an area of 25 km of Copenhagen University Hospital Hvidovre and Copenhagen University Hospital Bispebjerg for both cohorts [6]. All patients gave written and verbal informed consent prior to this study. PAL was measured around the clock for three weekdays and Saturday and Sunday. Patients who had sensor recorded PAL data for at least three valid days (72 h) and complete data on Hospital Anxiety and Depression Scale (HADS-A and HADS-D) were included in this exploratory study.

### 2.2. Assessments

All assessors completed a four-hour assessment course to ensure identical and standardized protocol procedure, including programming of sensors and recording of results. In addition, assessors observed at least four live assessments before being accredited as an assessor. The administration procedure reflects the conditions in everyday clinical practice, where several performance tests and questionnaires are conducted within a narrow time frame. The assessment procedures were reproducible and have been published. The interclass correlation consistency ranges from 0.86 to 0.98, with more details on absolute measurements in the publication [29,30].

Physical activity level and sleep were objectively measured with the AP triaxial accelerometer (PAL Technologies Ltd., Glasgow, UK). The AP is a device worn on the front thigh that gives detailed information about dynamic and static acceleration and can discriminate body posture as lying and sleeping, lying and awake, sitting, standing, stepping, cycling and seated transport [31]. The AP makes measurements of positions every 20th of a second. This raw acceleration and position data is packaged and stored in memory. The recording is uploaded from a docking station to the software program PALanalysis which is an automated algorithm. AP is a valid instrument to measure the number of steps per day in patients with COPD [32,33]. The variable steps per day was dichotomized to <5000 and ≥5000, based on prior research on older adults and patients with COPD, which shows that PAL of less than 5000 steps per day is detrimental and is thus defined as inactivity [34].

Symptoms of anxiety and depression were collected at baseline with the validated questionnaire “The Hospital Anxiety and Depression Scale” (HADS). The questionnaire consists of 7 questions for depression and anxiety and takes approximately 2–5 min to fill out [35]. HADS is widely recognized as a valid method for measuring symptoms of anxiety and depression in patients with COPD [36,37], while disagreement about which cutoff scores should be used as discriminative to determine the presence of symptoms exists [38,39]. We used the following cut-off scores for the analysis, based on the literature: a score of ≤7 indicates no presence of symptoms of depression or anxiety, a score of 8–10 indicates mild symptoms, 11–14 moderate symptoms and 15–21 severe symptoms of depression/anxiety [35]. Finally, patients were dichotomized into having “no symptoms” or “symptoms”, where the three levels of mild, moderate and severe symptoms were combined into one group with “symptoms”.

Time in bed (TIB) and sleep disturbances/Numbers of nocturnal sleeping bouts (NSB) were derived from the timecoding by the AP as total minutes lying down in bed and number of changes in posture from lying to sitting at night, using the default algorithm of the device. Cut-off scores for TIB and NSB were defined as: ≤9 h and ≤4 NSB, respectively [12,40,41]. Metabolic equivalents (METS) is a simple method to express energy cost of physical activity [42]. We distinguish light activity from sedentary behavior with a cut-off score defined as “any waking behavior characterized by an energy expenditure ≤ 1.5 METs while in a sitting or reclining posture” [43].

Data from AP on PAL and sleep was exported through the software program PALanalysis to Excel. The raw data was sorted in a meaningful order and screened for any errors before being exported to the statistical software package.

Descriptive and other variables included Body Mass Index, smoking status, Charlson Comorbidity Index, spirometry, 6-Minutes Walking Distance (6MWD) and anthropometric measures, were defined according to the standardized protocols from the Danish Society of Respiratory Medicine and international technical guidelines for walking test [44,45].

### 2.3. Analysis and Statistics

Descriptive data are presented as means with standard deviation (SD) for continuous data and as medians (25th, 75th percentile) or number and percentage for ordinal data not normally distributed. Difference between those categorized as physically inactive and those categorized as physically active were either analyzed with a two-sample *t*-test or the Mann–Whitney U test where assumptions were not satisfied (e.g., skewed data). The variable average number of steps per day was analyzed with negative binomial regression with a dispersion parameter to account for over dispersion, since the distribution of number of steps per day were skewed. An overall adjusted analysis was performed, adjusting for the confounders *age, gender, BMI, civil status/marital status, educational level, smoking, lung function, comorbidities and walking capacity* and results presented as rate ratio (RR) and confidence intervals (95% CI). Data were analyzed using IBM SPSS Statistics 21.0 (IBM, Chicago, IL, USA) and a *p*-value < 0.05 was considered significant.

## 3. Results

### 3.1. Daily Physical Activity in COPD

In total, 148 patients belonging to spirometric GOLD grades 3 and 4 (97.3%) with a mean ± SD FEV_1_% predicted of 33.6 ± 10.9% were eligible for analysis. Patients walked a median (25th, 75th percentile) of 2358 (1325.75; 3822.25) steps per day, with a total of 130 patients completing <5000 steps per day. Patients used the activity monitor for a mean time of 3 ± 0.5 weekdays, 1.7 ± 0.5 weekend days and a total of 4.7 ± 0.6 days. Demographic characteristics are shown in Table 1. A statistically significant proportion of the patients in the sedentary group (48.4% vs. 5.9%) used walking aids, had lower FEV_1_%predicted compared to the active group, shorter walking distance on 6MWD and fewer Sit-To-Stand (STS) repetitions compared to the group of more active patients (Table 1). Patients in the sedentary group walked significantly fewer steps per day median (25th, 75th percentile) of 2074.5 (1202.3; 3109.8) vs. 6605.5 (5393.3; 8705) (*p* < 0.001; Mann–Whitney U effect size r = 0.56), had significant shorter stepping time effect size of (*p* < 0.001; Mann–Whitney U effect size r = 0.56)) and spent significantly more time sitting (*p* < 0.004; Mann–Whitney U effect size r = 0.24) compared to the active group of patients (Table 2). None of the participants had any walking bouts > 20 min during the AP wearing time. Patients in both groups had a mean metabolic equivalent of task (METs)/hour of 1.3 and 1.4, corresponding to sedentary to light activity intensity levels on a day (24-h). The walking capacity (6MWD) was associated with steps per day (RR 1.004, 95%CI [1.003; 1.005]) (*p* < 0.001), as was severity of airway obstruction (FEV_1_%-expected) (RR 1.012, 95%CI [1.002; 1.022]) (*p* = 0.014) and an increased BMI (RR 1.021, 95%CI [1.004; 1.039]) (*p* = 0.014) (Table 3).

### 3.2. Association between Symptoms of Anxiety and Depression versus Daily Physical Activity

The patients were characterized by a median (25th, 75th percentile) HADS-A score of 5 (3; 8) and HADS-D score of 3 (2; 6), corresponding to no symptoms of anxiety or depression, respectively.

A higher percentage of active patients reported having symptoms of severe anxiety 22.2% versus 10.8% of sedentary patients. There was no significant difference in distribution of mild to severe symptoms of depression between the sedentary and active group of patients.

The adjusted negative binomial regression revealed that each one-point increase in anxiety score on the HADS-A, corresponded to a significant increase of 3.7% steps per day (RR 1.037, 95%CI [1.005; 1.070]) (*p* = 0.024) while each one-point increase in depression score on the HADS-D, corresponded to a significant decrease of 5% steps per day (RR 0.950, 95%CI [0.914; 0.986]) (*p* = 0.008).

### 3.3. Association between Objective Measured Sleep versus Daily Physical Activity

Patients spent a median (25th, 75th percentile) of 8.3 (7.1; 9.7) hours/night in bed and experienced 1.5 (0.8; 3) numbers of NSB. There was no significant difference between the two activity groups in distribution of TIB or NSB. Overall, 21% of the patients stayed < 7 h in bed and neither TIB (RR 0.966, 95%CI [0.920; 1.016]) (*p* = 0.179) nor NSB (RR 1.020, 95%CI [0.970; 1.072]) (*p* = 0.445) were significantly associated with steps per day in this sample of patients (Table 3).

## 4. Discussion

The main findings from this exploratory retrospective, cross-sectional study was that anxiety was associated with an increase in PAL, whereas the opposite was found regarding depression. TIB and NSB were not significantly associated with PAL. To the best of our knowledge this is the first study to combine and present novel data on the association between objectively monitored physical activity and sleep-measures and self-reported symptoms of depression and anxiety, in patients with severely progressed COPD.

Nguyen et.al. (2013) performed a cross-sectional study which aimed to establish the association between symptoms of anxiety and depression and physical activity in 148 patients with stable COPD [14].As in our study, the authors found the same trend with an increase in the number of steps with an increase in HADS-A and decreasing number of steps with an increase in HADS-D. Patients had their PAL measured using the validated activity monitor Stepwatch 3. Their results showed that each 1-point increase in HADS-A was associated with an increase of 288 steps per day (SE: 80), (*p* < 0.01). Furthermore, they found that per each 1-point increase in HADS-D, patients walked 176 fewer steps per day (SE: 76), (*p* = 0.02) [14]. By contrast, a cross-sectional study by Tödt et al. from 2015 could not establish the same association between symptoms of anxiety and an increased PAL [46]. Tödt et.al. 2015 examined which factors were associated with a low PAL in patients with stable COPD (N = 101, FEV_1_ 50%, Age 68, 6MWD 335 to 461m, HADS A/D, median 4/5), using the International Physical Activity Questionnaire (IPAQ-S) [46]. They found no differences between patients with low, moderate, and high self-reported physical activity, in relation to anxiety or depressive symptoms measured by HADS.

Some mechanisms have been proposed to support the fact that patients diagnosed with anxiety are characterized by psychomotor hyperactivity [47], and similarly, patients diagnosed with depression have reduced physical activity, e.g., on the background of a general low mood [48]. Thus, similar mechanisms are plausible in patients with COPD with symptoms of depression and/or anxiety. It is thus possible that the increased amount of psychomotor hyperactivity in patients with anxiety may create an unintentional behavioral change in the form of involuntary restlessness and general increased level of physical activity [20,47,48]. Our sample was characterized by having a low mean HADS-A score 5.6 (3.7) and thus it plausible that this association would be J- or U-shaped if a patient reaches high anxiety levels where anxiety likely would impair their overall behavior more negatively including the level of physical activity. A plausible threshold was however not possible to test in this study due to the HADS score distribution in our cohort. Conversely, a general low mood that is present in patients experiencing depression has an opposite behavioral change in the form of decreased physical activity [16,22,48,49]. It may thereby be the underlying mechanisms that influence the results of our study and Nguyen et al., both of which point in the direction that the dominant symptoms of depression and anxiety have different effects on physical activity. A plausible explanation for the finding by Tödt el al. could be that the use of self-reported physical activity compares poorly to objectively measured data and furthermore tends to misclassify physical activity [49,50]. Furthermore, it would have been interesting to compare our results with a control group, to identify if increasing PAL can alleviate symptoms of anxiety and depression. This was unfortunately not possible due to the nature of our retrospective cross-sectional design. Despite of this we are familiar with a Swedish longitudinal study based on health care workers concluding that PAL were associated with symptoms of anxiety and depression across time, measured with HADS [51].

### 4.1. Time in Bed and Nocturnal Sleeping Bouts

There is only modest documentation on symptoms of sleep disturbances in patients with COPD and it is on top of that an overlooked phenomenon in the treatment of COPD [52]. Furthermore, OSA and poor sleep are commonly experienced symptoms in patients with COPD [53]. A cross-sectional study by Spina et al. from 2016 [12], aimed to investigate the association between TIB and NSB and next-day physical activity (number of steps per day) in patients with COPD (N = 932), with a FEV1% predicted mean 50.8 (SD: 20.5). Data on physical activity and sleep were measured with the activity monitor SenseWear bracelet or mini bracelet (location: upper arm) [12]. In contrast to our findings, the study found the following trend between increased TIB and reduced number of steps per day, in their study sample: TIB <347 min = 4952 steps, TIB 347–416 min = 4882 steps, at TIB 416–480 min = 4759 steps and at TIB ≥480 min = 4503 steps, (*p* = 0.08) [12]. The study also found that patients with COPD walked 5136 steps at <2 nocturnal sleeping bouts (NSB), 4874 steps at 2–3 NSB, 4664 steps at 3–4 NSB and 4484 steps at ≥4 NSB, (*p* = 0.08), thus a trend toward an increase of NSB associate to a decrease in steps per day [12]. We did not find any trends or associations between NSB and number of steps per day (*p* = 0.445).

In comparison, Hirata et al. in 2020 [13] aimed to investigate the association between sleep and physical activity in patients with COPD (N = 55, FEV1 55%, Age 67, 6MWD 473 m, PAL step per day 6070, HADS A/D 5/3). Data on physical activity and sleep were measured with the SenseWear Pro2 (location: upper arm) [13]. The study found that ≤7, 7–9, and ≥9 h of TIB were associated with a median value of 8945 (IQR: 6746; 11,956), 8424 (IQR: 4068; 10,481) and 3018 (IQR: 1768; 4586) steps per day, respectively, (*p* < 0.01) [13]. In contrast, our study failed to establish a statistically significant association between TIB and steps per day (*p* = 0.642). Furthermore, the study found that <6.9 and >6.9 numbers of sleep disturbances were associated with median values of 8813 (IQR: 3554; 10,951) and 5914 (IQR 3452; 9348) steps per day, respectively (*p* = 0.439) [13]. In contrast to our study, both Spina et al. and Hirata et al. [12,13] found a trend for sleep interruptions to be associated with fewer steps per day. However, it must be noted that Spina et al. only included those patients with COPD who had >4 h of bedtime per day and neither study could establish a significant association between NSB and steps per day. A plausible explanation for the inconsistent results could be that the activity monitor SenseWear Pro2 and AP has a predefined algorithm for bed-time and wake-time, which has a great deal of influence on whether data is recorded as sleep or just rest, and thus whether data is interpreted correctly [13].

Another explanation could be that patients with COPD are more exposed to nocturnal hypoxemia, which leads to more position changes and thereby increased nocturnal activity to improve their ventilation^24^. However, nocturnal hypoxemia is not distinctive to COPD, whereas OSA may be the leading cause of sleep disturbances [23].

In summary, there are several theoretical mechanisms that could explain how sleep disturbances has a worsening impact on PAL in COPD. It is known that sleep disturbances is more prominent in patients with severe airflow limitation and in those with nocturnal dyspnea, especially patients suffering from OSA [23,53]. Moreover, sleep disturbances seem to be associated with the ability to engage in physical activity as well as the capability to complete a PR course for patients with COPD [54]. Considering the novel and conflicting findings from this study, much more research is needed to identify a potential causal association and impact between sleep disturbances and physical activity in patients with COPD and various intra- and extrapulmonary manifestations.

### 4.2. Strengths and Limitations

Physical activity and sleep were measured with an activity monitor which is considered the best method for accurate measurements of movements and physical activity. The outcome variables were measured around the clock for 5 days including weekends which we believe reflects a more valid expression of physical activity and sleep patterns [55]. Furthermore, it is a strength that AP is a validated method for measuring steps per day in patients with COPD [32,33]. AP has a precision of steps down to 1.5 miles/h (2.41 km/h) [56], which is an advantage in avoiding underestimation of physical activity in a patient with COPD, as slow gait due to high symptom burden is often seen [57]. The primary limitation of our cross-sectional study design is that exposure and outcome are simultaneously assessed, and thus it is not possible to establish a cause-and-effect relationship. A limitation that also needs to be considered, is the circumstance that there are no known activity sensors validated for measuring and recording the quantity and quality of sleep, which may explain the directional variation across studies. We also acknowledge that neither TIB or NSB fully characterize the measurement of sleep quality and sleep duration whereas, e.g., sleep onset latency and wake onset would also be of interest. Regrettably, the AP sensor is not able to generate further descriptive variables on sleep. We recruited patients with identical real-world inclusion criteria for hospital-based PR. This group may present different activity and symptom pattern in contrast to those with milder symptoms followed by their general practitioners and receiving community based PR. Data on OSA was not available in our study sample which is a limitation [23]. The ability to include patients with pronounced anxiety or depression symptoms is in general difficult to recruit into clinical studies, and thus an important focus moving forward with research in respiratory medicine in general.

## 5. Conclusions

In this cross-sectional study of patients with severe COPD, increase in HADS-A score was associated with a significant increase in steps per day while increase in HADS-D score was associated with a significant decrease in steps per day. Time in bed and nocturnal bouts was not significantly associated with PAL in these patients.

## Figures and Tables

**Table 1 ijerph-19-16804-t001:** Characteristics of included patients.

Variables	All (n = 148)	Sedentary <5000 Steps per Day (n = 130)	Physical Active ≥5000 Steps per Day (n = 18)
Female sex, n (%)	79 (51.6)	69 (53.1)	10 (55.6)
Age, yr	69.77 ± 8.4	70.43 ± 8.2	65 ± 8.5 *
Educational level yr, n (%) <12 ≥12	114 (74.5) 34 (25.5)	104 (80) 26 (20)	10 (55.6) 8 (44.4)
Marital status, n (%) Married/living with partner Living alone	107 (72.3) 41 (27.7)	93 (71.5) 37 (28.5)	14 (77.8) 4 (22.2)
Body mass index, kg/m^2^	25.7 ± 5.9	25.88 ± 6	24.19 ± 4.9
FEV_1_, % predicted	33.6 ± 10.9	32.3 ± 10.8	42.9 ± 7.5 *
FEV_1_/FVC, %	42.9 ± 12.2	42.4 ± 12.1	46.9 ± 12 *
GOLD I/II/III/IV, n (%)	0/4 (2.7)/82 (55.4)/62 (41.9)	2.3/51.5/46.2	5.6/83.3/11.1 *
A/B/C/D, n %	1 (0.7)/69 (46.6)/5 (3.4)/73 (49.3)	0.8/48.5/2.3/48.5	0/33.3/11.1/55.6
LTOT, n (%)	22 (15)	21 (16.2)	1 (5.6)
SpO_2_ at rest (%)	94.6 (2.7)	94.4 (2.8)	95.7 (1.8)
MRC score, median (25th, 75th percentile)	4 (3; 4)	4 (3; 4)	3 (3; 3) *
Smoking status, n (%) Never Former Current Pack-year history	2 (1.4) 116 (78.4) 30 (20.3) 45.9 (21.3)	1 (0.8) 104 (80) 25 (19.2) 46.9 (22.1)	1 (5.6) 12 (66.7) 5 (27.8) 39.3 (13.5)
BODE index points, median (25th, 75th percentile)	5 (4; 7)	6 (4; 7)	3 (2.8; 4) *
Number of comorbidities, median (25th, 75th percentile)	2 (1; 3)	2 (1; 3)	2 (1; 3)
Exacerbations, previous 12-month, median, (25th, 75th percentile)	0 (0; 6)	0 (0; 2)	1 (0; 2)
Walking aid, Yes, n (%)	63 (43.4)	62 (48.4)	1 (5.9) *
6MWD, meters #	327 ± 122.4	302.1 ± 111.8	466.9 ± 75.2 *
30-sec-STS, repetitions	9.7 ± 5.5	9 ± 4.9	14.3 ± 7.7 *
CAT, score points	19.9 ± 6.6	20.2 ± 6.4	17.3 ± 7.8
HADS-Anxiety n (%) No symptoms Mild—Moderate symptoms Severe symptoms HADS-Depression n (%) No symptoms Mild—Moderate symptoms Severe symptoms HADS-Anxiety score, median (25th, 75th percentile) HADS-Depression score, median (25th, 75th percentile)	105 (70.9) 27 (18.2) 16 (10.8) 126 (85.1) 17 (11.5) 5 (3.4) 5 (3; 8) 3 (2; 6)	92 (69.2) 26 (20) 12 (10.8) 110 (84.6) 15 (11.5) 5 (3.9) 5 (3; 8) 3 (2; 6)	13 (72.2) 1 (5.6) 4 (22.2) 16 (88.9) 2 (11.1) 0 4 (3; 9) 2 (1; 4.25)
EQ-5D, VAS score EQ-5D, index score	54.5 ± 20 0.7 ± 0.1	54.17 ± 20.3 0.7 ± 0.1	57.17 ± 17.9 0.7 ± 0.2

Data are presented as mean ± SD except where otherwise indicated. Any statistically significant difference between sedentary <5000 steps per day and physical active ≥5000 steps per day is denoted * *p* < 0.05. # in total 103 patients completed the 6MWD. Data on pack-year history was available for 139 patients. *Definition of abbreviations:* FEV_1_, forced expiratory volume in the first second; FVC, forced vital capacity; GOLD, Global initiative for Chronic Obstructive Lung Disease; A/B/C/D, risk stratification; LTOT, Long-term oxygen therapy; SpO_2_, arterial oxygen saturation as measured by pulse oximetry; MRC, Medical Research Council; BODE index, body mass index, airflow obstruction, dyspnea, and exercise capacity; CAT, COPD Assessment Test; HADS, hospital anxiety and depression scale; EQ-5D, Euro-Qol dimension.

**Table 2 ijerph-19-16804-t002:** Characteristics of physical activity and sleep measurements.

Variables	All (n = 148)	Sedentary <5000 Steps per Day (n = 130)	Physical Active ≥5000 Steps per Day (n = 18)
Number of steps per day, median (25th, 75th percentile) Number of steps < 1 min Number of steps 1 to 5 min Number of steps > 5 to 10 min Number of steps > 10 to 20 min Number of steps > 20 min	2358.5 (1325.75; 3832.25) 1851.5(1191.3; 2817.5) 242 (32.5; 581.3) 0 (0; 61.5) 0 (0; 0) 0 (0; 0)	2074.5 (1202.3; 3109.8) 1699.5(1130; 2431) 141 (141; 485) 0 (0; 0) 0 (0; 0) 0 (0; 0)	6605.5 (5393.3; 8705) * 4144 (3584.3; 5693) * 1841 (1397; 2193.5) * 188 (94.5; 356.8) * 0 (0; 51.8) * 0 (0; 0)
Total stepping time, minutes Stepping time < 1 min Stepping time 1 to 5 min Stepping time > 5 to 10 min Stepping time > 10 to 20 min Stepping time > 20 min Total standing time, minutes, median (25th, 75th percentile) Total sitting time, minutes, median (25th, 75th percentile)	32.5 (19.7; 52.1) 27.7 (17.8; 40.3) 2.9 (0.5; 7.1) 0 (0; 0.8) 0 (0; 0) 0 (0; 0) 151.9 (92.3; 233.9) 650.2 (524.5; 782.4)	29.6 (17.7; 46.2) 24.7 (17.1; 35.8) 1.9 (0.3; 5.6) 0 (0; 0) 0 (0; 0) 0 (0; 0) 138.3 (86.95) 668.5 (537.3; 788.4)	89 (73.1; 105.9) * 59.6 (51.7; 81.5) * 20.7 (14.8; 22.3) * 1.9 (1; 3.5) * 0 (0; 0.5) * 0 (0; 0) 280.4 (197.8; 323.7) * 530.9 (467.99; 636.99) *
Number of sit to stand transitions, repetitions	37.1 ± 15.1	35.6 ± 14.5	48.1 ± 15.5 *
METS/hour, METs	1.3 ± 0.1	1.3 ± 0.1	1.4 ± 0.1 *
TIB, minutes, median (25th, 75th percentile) <7 h, n (%) 7–9 h n (%) >9 h n (%) NSB, numbers, median (25th, 75th percentile)	8.3 (7.1; 9.7) 32 (21.6) 67 (45.3) 49 (33.1) 1.5 (0.8; 3.0)	8.3 (7.1; 9.8) 28 (21.6) 57 (43.8) 45 (34.6) 1.6 (0.8; 3.2)	8.3 (7.2; 9.0) 4 (22.2) 10 (55.6) 4 (22.2) 1.3 (1; 2.3)
Number of days worn, mean ± SD Number of worn weekdays, mean ± SD Number of worn weekend days, mean ± SD	4.7 ± 0.6 3 ± 0.5 1.7 ± 0.5	4.72 ± 0.6 3 ± 0.5 1.7 ± 0.5	4.8 ± 0.5 3.17 ± 0.5 1.7 ± 0.5

Data are presented as median (25th, 75th percentile) and mean ± SD per day. Any statistically significant difference between Sedentary <5000 steps per day and physical active ≥5000 steps per day is denoted * *p* < 0.05. *Definition of abbreviations:* METs, metabolic equivalents; TIB, time in bed duration; NSB, number of nocturnal sleeping bouts.

**Table 3 ijerph-19-16804-t003:** Negative binomial regression and rate ratio for number of steps per day.

	RR (95%CI)	*p*-Value
Gender (ref. female)	1.079 (0.900; 1.295)	0.412
Civil status (ref. living alone)	1.117 (0.913; 1.366)	0.282
Walking aid (ref. no aid)	0.925 (0.730; 1.173)	0.521
Age (years)	1.009 (0.996; 1.022)	0.179
Education (ref. ≥ 12 yr)	0.962 (0.785; 1.177)	0.705
FEV_1_% expected	1.012 (1.002; 1.022)	0.014
BMI (kg/m2)	1.021 (1.004; 1.039)	0.014
Smoking status (ref. no smoking)	0.961 (0.757; 1.221)	0.746
Comorbidity (numeric)	0.966 (0.902; 1.035)	0.325
6MWD (meters)	1.004 (1.003; 1.005)	0.001
HADS-A (points)	1.037 (1.005; 1.070)	0.024
HADS-D (points)	0.950 (0.914; 0.986)	0.008
TIB (hours)	0.966 (0.920; 1.016)	0.179
NSB (numbers)	1.020 (0.970; 1.072)	0.445

Data are presented as rate ratio and 95% confidence interval. *Definition of abbreviations:* FEV_1_, forced expiratory volume in the first second; BMI, body mass index; 6MWD, 6 min walk distance; HADS, hospital anxiety and depression scale; TIB, time in bed duration; NSB, number of nocturnal sleeping bouts.

## Data Availability

The raw data are not available to the public according to rules of the Danish Data Protection Agency.

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
