# Peer review of "Sleep Quality and Self-Reported Symptoms of Anxiety and Depression Are Associated with Physical Activity in Patients with Severe COPD"

_ijerph, 2022, doi:10.3390/ijerph192416804_

Round 1

Reviewer 1 Report

The manuscript describes the association of sleep quality and quantity and  self reported depresion and anxiety indexes with level physical activity in COPD patients, and found statisticaly significant association between anxiety and depresion scores with number of steps per day in patients with severe COPD. The study is interesting and well writen, limitations and strenghts alisted, but the authors should also include significance and novelty of the study. Also, tables seem pretty cluttered and the results should be better presented.

Author Response

Review Report Form 1

Comments and Suggestions for Authors

The manuscript describes the association of sleep quality and quantity and self reported depresion and anxiety indexes with level physical activity in COPD patients, and found statisticaly significant association between anxiety and depresion scores with number of steps per day in patients with severe COPD. The study is interesting and well writen, limitations and strenghts alisted, but the authors should also include significance and novelty of the study.

Dear reviewer, we appreciate that you find our study interesting and comprehensible. Regarding significance and novelty of the study. We have added following under discussion: “To the best of our knowledge this is the first study to combine and present novel data on the association between objectively monitored physical activity and sleep-measures and self-reported symptoms of depression and anxiety, in patients with severely progressed COPD” (please read: p.6, line 201-203).

Also, tables seem pretty cluttered and the results should be better presented.

Thank you for emphasizing that you find the tables cluttered. We agree that they contain lots of numbers. We have increased the section and line space to make the table less cluttered. Furthermore, we have changed the layout slightly which we hope meets your raised concern, however we find that the content of the tables is appropriate, and we have thus not omitted any variables. We are happy to receive more specific suggestions for table changes (please read table 1 to 3).

Reviewer 2 Report

In this study, the authors aimed to examine the association between sleep, anxiety, depression, and physical activity in patients with severe Chronic Obstructive Pulmonary Disease (COPD). Overall, the article is well-written and comprehensive. However, as explained below, I raised some points that need to be addressed

Major points:

- The authors should give more information on data acquisition and algorithms used to measure physical activity and sleep.

- Is the used accelerometer validated for sleep assessment? please provide a reference

- In my opinion time in bed (TIB) and Numbers of nocturnal sleeping bouts (NSB) are insufficient to characterize sleep, an essential variable of this study. Is it possible to retrieve common sleep outcomes such as sleep efficiency, sleep onset latency, and wake after sleep onset?

Minor points:

-       Line 17: Please define COPD in its first appearance in the text.

-       Line 90-92: Please specify, do you mean among COPD patients?

Author Response

Review Report Form 2

In this study, the authors aimed to examine the association between sleep, anxiety, depression, and physical activity in patients with severe Chronic Obstructive Pulmonary Disease (COPD). Overall, the article is well-written and comprehensive. However, as explained below, I raised some points that need to be addressed

Major points:

- The authors should give more information on data acquisition and algorithms used to measure physical activity and sleep.

We have added additional information on the data acquisition in the manuscript (please read p. 4, line 109-113).

ActivPAL sensors are validated to measure physical activity level including in patients with COPD
(Cindy Ng LW, Jenkins S, Hill K. Accuracy and responsiveness of the stepwatch activity monitor and ActivPAL in patients with COPD when walking with and without a rollator. Disabil Rehabil. 2012;34(15):1317-1322.)

Furthermore, in comparative studies, activPAL have proven to measure equally to other commonly used fabric devices (Sensewear, McRoberts, Actigraph). (
Storm FA, Heller BW, Mazzà C (2015) Step Detection and Activity Recognition Accuracy of Seven Physical Activity Monitors. PLoS ONE 10(3): e0118723.) doi:10.1371/journal.pone.0118723.

To our knowledge, the algorithms for measuring physical activity and sleep has not been published in detail for the most commonly used accelerometer-based devices (SENS, Sensewear, McRoberts, Actigraph, ActivPAL). Thus, it’s difficult to describe the algorithm and the underlying mathematics and physics applied to the different devices.

Before we started our analysis, we’ve made several attempts to contact the company PAL technologies Ltd. regarding specific information on the sleep measure algorithm of the device without getting a reply.

Thus, we are not able to present more information than added in the manuscript. (please read p. 4, line 118-121).

- Is the used accelerometer validated for sleep assessment? please provide a reference

This is a central issue moving forward with research and measurement of various sleep measures. To our knowledge, no known accelerometers are validated to measure sleep and various sleep measures. We have addressed this central issue in our discussion-section (p. 9, line 292-294). Naturally this is a limitation throughout the published literature. We have several days of assessment with the monitor which should reduce the implicit measurement errors in the activPAL and thus increase the reliability of our data.

From one of our cohorts (NCT04249388) we have collected data from both ActivPAL sleep measures and Pittsburgh Sleep Quality Index (PSQI) (n=76). We have performed identical negative binomial regression were TIB measured with ActivPAL was substituted with self-reported PSQI variables time in bed and hours of sleep per night on at the time. Furthermore, we also performed the analysis with the calculated PSQI domain (Sleep quality, sleep latency, sleep duration, habitual sleep efficiency, sleep disturbance, daytime dysfunction and PSQI total score) one at the time. None of these variables were significant nor did they change the primary significant findings from HADS-A, HADS-D, 6MWD, BMI and FEV1. Thus, indicating similar findings from the ActivPAL variables as to self-reported sleep quality questionnaire.

- In my opinion time in bed (TIB) and Numbers of nocturnal sleeping bouts (NSB) are insufficient to characterize sleep, an essential variable of this study. Is it possible to retrieve common sleep outcomes such as sleep efficiency, sleep onset latency, and wake after sleep onset?

Thank you for your opinion. We recognize that our variables only partially characterize various sleep measures of sleep quality in patients with COPD. We agree that sleep outcomes such as sleep efficiency, sleep onset latency, and wake after sleep onset is of interest. Unfortunately, the AP sensor are not able to generate further descriptive sleep variables, than TIB and NSB.
As we presented above, Sleep quality, sleep latency, sleep duration, habitual sleep efficiency, sleep disturbance, daytime dysfunction and PSQI total score from Pittsburgh Sleep Quality Index did not change the main findings in the regression model. We have no knowledge of any other activity monitors that are gold-standard validated to present sleep variables whether it concerns time in bed, number of nocturnal sleeping bouts, sleep efficiency, sleep onset latency, and wake after sleep onset. Please read our strength and limitation section where we have added further limitation of sleep measurements, p.9 line 294-296. 

Minor points:

-       Line 17: Please define COPD in its first appearance in the text.

Thank you for addressing this point. We have corrected this in the manuscript, (please read p. 1 line 16-17).

-       Line 90-92: Please specify, do you mean among COPD patients?

We were not able to identify the sentence mentioned above. We found a similar sentence in p. 8, line 242 and changed the wording, based on your comment.

Reviewer 3 Report

Thank you for inviting me to review the article entitled “Sleep quality and self-reported symptoms of anxiety and depression are associated with physical activity in patients with severe COPD.”

This article identified a statistically significant association between Anxiety (HADS-A) and depression (HADS-D) with a series of daily steps in patients with severe COPD from 148 participants with COPD. Studies have shown that an increased HADS-A score is associated with a significantly higher daily step count. In comparison, an increased HADS-D score is associated with a significantly lower daily step count. Sleep and nocturnal seizures were not significantly associated with daily physical activity (PAL) in these patients.

Overall, the article is informatively organized with fluent expression and logical coherence. These minor revisions need to be considered before publication:

1.      The paper cited an article titled “Patients with COPD with higher levels of anxiety are more physically active,” which believed that increased physical activity in COPD patients was related to anxiety, and the two articles were partly consistent in their investigation methods and conclusions. Moreover, this paper adds that the reduction of PAL is related to depression; PAL and sleep time are unrelated. Unfortunately, the innovation of this paper is a little lacking.

2.      Is the conclusion that PAL is associated with HADS-A and HADS-D specific to COPD patients? Is PAL also associated with HADS-A and HADS-D in non-COPD patients?

3.      Among those methods, a critical approach was applied to test sleep quantity and quality. The total number of minutes the patient lies in bed and the change in position from lying to sitting during the night are inaccurate enough to record. Is there a more accurate detection method that can be applied here?

4.      Is there a need for a control group with increased PAL that should be added to the article? And the HADS tests were performed on the control group to verify further that increasing PAL can alleviate depression and anxiety.

Author Response

Review Report Form 3

Comments and Suggestions for Authors

Thank you for inviting me to review the article entitled “Sleep quality and self-reported symptoms of anxiety and depression are associated with physical activity in patients with severe COPD.”

This article identified a statistically significant association between Anxiety (HADS-A) and depression (HADS-D) with a series of daily steps in patients with severe COPD from 148 participants with COPD. Studies have shown that an increased HADS-A score is associated with a significantly higher daily step count. In comparison, an increased HADS-D score is associated with a significantly lower daily step count. Sleep and nocturnal seizures were not significantly associated with daily physical activity (PAL) in these patients.

Overall, the article is informatively organized with fluent expression and logical coherence. These minor revisions need to be considered before publication:

Dear reviewer. Thank you for your acknowledgment. We are very pleased that you find our article informative and interesting.

  1. The paper cited an article titled “Patients with COPD with higher levels of anxiety are more physically active,” which believed that increased physical activity in COPD patients was related to anxiety, and the two articles were partly consistent in their investigation methods and conclusions. Moreover, this paper adds that the reduction of PAL is related to depression; PAL and sleep time are unrelated. Unfortunately, the innovation of this paper is a little lacking.

Dear reviewer. Thank you for this valid point.

We agree that the area has been researched and some  of our results are consistent with previous findings and thus not novel findings per se. Respectfully, it is our opinion that one, two or three cross-sectional studies in various selected patient populations hardly are saturated evidence for making a firm case of hypothetical causality before moving forward with longitudinal studies, with larger samples, control groups and expensive objective assessment of sleep (polysomnography/polygraphy). We do believe that our study ads to the growing evidence as we only included patients with severe progressed disease burden and airway obstruction, which is not the case in any of the other studies. Lastly, this is the first study to investigate the collective association and impact between objectively monitored physical activity and both sleep-measures and self-reported symptoms of depression and anxiety.

  1. Is the conclusion that PAL is associated with HADS-A and HADS-D specific to COPD patients? Is PAL also associated with HADS-A and HADS-D in non-COPD patients?

Yes, HADS-A and HADS-D was statically associated to number of steps per day in patients with severe to very severe COPD. We do not have data from a matched control group without COPD and not able to answer the second question from our data. We are familiar with a Swedish longitudinal population-based study in healthcare workers concluding that changes in physical activity were associated with changes in depression, anxiety, and burnout across time when measured by HADS-A and HADS-D. (Lindwall, M., Gerber, M., Jonsdottir, I. H., Börjesson, M., & Ahlborg, G., Jr. (2014). The relationships of change in physical activity with change in depression, anxiety, and burnout: A longitudinal study of Swedish healthcare workers. Health Psychology, 33(11), 1309–1318.).

We have added additional information in our discussion section. (Please read p. 7 line 232-237).

  1. Among those methods, a critical approach was applied to test sleep quantity and quality. The total number of minutes the patient lies in bed and the change in position from lying to sitting during the night are inaccurate enough to record. Is there a more accurate detection method that can be applied here?

To our knowledge, no known accelerometers (SENS, Sensewear, McRoberts, Actigraph, ActivPAL) are validated to measure sleep and various sleep measures. As replied to reviewer nr. 1, we have addressed this central issue in our discussion-section (p. 9., line 292-294). Naturally this is a limitation throughout the published literature. Unfortunately, the ActivPAL software algorithm only generate time in bed and number of nocturnal sleeping bouts.
Sleep quality, sleep latency, sleep duration, habitual sleep efficiency, sleep disturbance, daytime dysfunction and PSQI total score from Pittsburgh Sleep Quality Index did not change the main findings in the primary regression model presented in this manuscript.

  1. Is there a need for a control group with increased PAL that should be added to the article? And the HADS tests were performed on the control group to verify further that increasing PAL can alleviate depression and anxiety.

We agree that the use of a control group would strengthen the association. This was unfortunately not a possibility in our study, due to the explorative nature of our retrospective cross-sectional design. We would recommend this as an idea for a future study design, based on the results from our study. (Please read our discussion section p. 7, line 232-237)

Round 2

Reviewer 2 Report

The authors recognized that the instrument used to measure sleep (activPAL) is not validated. I'm therefore concerned about a large part of the reported results. I'm unfortunately not able to accept this paper for publication.

Author Response

Dear reviewer,

We have been very transparent of the existing activity/ accelerometry sensors are validated for measuring physical activity, while they at present are validated for sleep. We think this transparency are important. While not validated for sleep these sensors are assumed and accepted to give an estimate of selected variable of sleep patterns. While Polysomnography are GOLD standard, in research and especially in larger sample activity/ accelerometry sensors and self-reported sleep like Pittsburgh Sleep Quality index (PSQI) are used for feasibility considerations and costs.

Several studies reporting on sleep based on estimate assessment from activity/ accelerometry sensors are published.

When searching in pubmed in your specific journal IJERPH, our pubmed search revealed more than 20 publication within the past two years covering sleep reported from activity sensors such actigraph, Sensewear and ActivePAL (similar our sensor) which are not validated. (please read search from published articles in IJERPH - sent to editorial office)

Flagship journal as Thorax and ERJ, ERJ open also publish article on sleep from not validated sleep monitors as sensors actigraph, Sensewear and ActivePAL as they are “best” feasible estimates currently.

Thus, it beyond our understanding why those many studies have been accepted using invalid measures of sleep, while in this circumstance suggest to rejects our study on the exact same principles. It appears very inconsistent og subjective to us.

Even more since this is the only “one sentence” statement/objection from the reviewer meanwhile the two other reviewers were overall positive of our manuscript in general.

Respectfully, we find the "one" argument for rejection very inconsistent, based on previous publication acceptance, and we think that this really undermines the credibility and thoroughness that the public and we as researchers think that peer-review processes provides.

In this case it seems hasty, subjective, and very inconsistent with previous academic decisions in 2021 and 2022.

By making this case, it is our strong hope that you will reconsider your assessment.

We are also willing and able to include the additional data from our self-reported questionnaire on sleep (Pittsburgh Sleep Quality index (PSQI)) whereas IJERPH have accepted more than four studies the past two years.